# Contrast sensitivity reveals an oculomotor strategy for temporally encoding space

Antonino Casile[1,2,3]*, Jonathan D Victor[4,5], Michele Rucci[6,7]*

[1]Center for Translational Neurophysiology, Istituto Italiano di Tecnologia, Ferrara, Italy; [2]Center for Neuroscience and Cognitive Systems, Rovereto, Italy; [3]Department of Neurobiology, Harvard Medical School, Boston, United States; [4]Brain and Mind Research Institute, Weill Cornell Medical College, New York, United States; [5]Department of Neurology, Weill Cornell Medical College, New York, United States; [6]Brain and Cognitive Sciences, University of Rochester, Rochester, United States; [7]Center for Visual Science, University of Rochester, Rochester, United States

**Abstract** The contrast sensitivity function (CSF), how sensitivity varies with the frequency of the stimulus, is a fundamental assessment of visual performance. The CSF is generally assumed to be determined by low-level sensory processes. However, the spatial sensitivities of neurons in the early visual pathways, as measured in experiments with immobilized eyes, diverge from psychophysical CSF measurements in primates. Under natural viewing conditions, as in typical psychophysical measurements, humans continually move their eyes even when looking at a fixed point. Here, we show that the resulting transformation of the spatial scene into temporal modulations on the retina constitutes a processing stage that reconciles human CSF and the response characteristics of retinal ganglion cells under a broad range of conditions. Our findings suggest a fundamental integration between perception and action: eye movements work synergistically with the spatio-temporal sensitivities of retinal neurons to encode spatial information.

DOI: https://doi.org/10.7554/eLife.40924.001

*For correspondence:
antonino.casile@iit.it;
toninocasile@gmail.com (AC);
rucci.michele@gmail.com (MR)

**Competing interests:** The authors declare that no competing interests exist.

## Introduction

Contrast sensitivity, the ability to distinguish a patterned input from a uniform background, is one of the most important measures of visual function (*Robson, 1966*; *Campbell and Robson, 1968*; *De Valois et al., 1974*; *Owsley and sensitivity, 2003*). Elucidation of its underlying mechanisms is, thus, essential for understanding how the visual system operates both in health and disease.

It has long been established that sensitivity varies in a specific manner with the spatial frequency of the stimulus, yielding the so-called contrast sensitivity function (henceforth CSF). Under photopic conditions, the CSF measured with stationary gratings exhibits a well-known band-pass shape that typically peaks around 3–5 cycles/deg and sharply declines at higher and lower spatial frequencies. The mechanisms responsible for this dependence on spatial frequency are not fully understood. At high frequency, a decline in sensitivity is expected for several reasons, including the filtering of the eyes' optics (*Campbell and Green, 1965*) and the spatial limits in sampling imposed by the cone mosaic on the retina (*Hirsch and Miller, 1987*; *Rossi and Roorda, 2010*). At low frequencies, however, the reasons for a reduced sensitivity have remained less clear.

A popular theory directly links the low-frequency attenuation in visual sensitivity to the neural mechanisms of early visual encoding (*Atick and Redlich, 1990*; *Atick and Redlich, 1992*). Building on theories of efficient coding (*Barlow, 1961*), it has been argued that this attenuation reflects a

form of matching between the characteristics of the natural visual world and the response tuning of neurons in the retina: retinal ganglion cells (henceforth RGCs) respond less strongly at low spatial frequencies so as to counterbalance the spectral distribution of natural scenes. According to this proposal, this filtering eliminates part of the redundancy intrinsic in natural scenes and enables more efficient (i.e. more compact) visual representations.

Although very influential, this proposal conflicts with experimental data. Neurophysiological recordings have long shown that the way the responses of retinal ganglion cells vary with spatial frequency deviates sharply from the CSF. The CSF of macaques is very similar to that of humans (*De Valois et al., 1974*); yet neurons in the macaque retina respond much more strongly at low spatial frequencies than one would expect from behavioral measurements of the CSF (*Figure 1A*). This deviation cannot be reconciled with standard models of retinal ganglion cells. It persists even when one takes into account obvious differences in the stimuli often used in neurophysiological and behavioral measurements (i.e. drifting gratings vs. temporally modulated gratings), as well as the nonlinear attenuation in responsiveness at low spatial frequencies exhibited by some retinal ganglion cells (*Derrington and Lennie, 1984*; *Croner and Kaplan, 1995*; *Benardete and Kaplan, 1997a*). This mismatch between neuronal and behavioral sensitivity indicates that additional mechanisms contribute to the CSF.

A fundamental difference between neurophysiological and behavioral measurements of contrast sensitivity is the presence of eye movements in the latter. Under natural viewing conditions, humans and other primates incessantly move their eyes (*Kowler, 2011*; *Cherici et al., 2012*). Small movements, known as fixational eye movements (FEMs), occur, even when attempting to maintain steady gaze on a single point (*Figure 1B*). Although humans often tend to suppress saccades of all sizes, including microsaccades, during measurements of contrast sensitivity (*Mostofi et al., 2016*), ocular drift—the seemingly erratic motion in between saccades/microsaccades—keeps the stimulus on the retina always in motion and may cover an area as large as that of the foveola (*Rucci and Poletti, 2015a*). Critically, this retinal image motion is completely eliminated or markedly attenuated in many neurophysiological preparations, where the retina is studied in a dish, or eye muscles are paralyzed as a result of anesthesia and/or neuromuscular blockade.

In previous work, we have shown that eye drift profoundly reshapes visual input signals, redistributing the 0 Hz (DC) power of the external static stimulus to non-zero temporal frequencies on the retina (*Casile and Rucci, 2006*; *Casile and Rucci, 2009*; *Kuang et al., 2012*; *Aytekin et al., 2014*). These modulations appear to be used by humans for the fine spatial discrimination (*Rucci et al., 2007*; *Boi et al., 2017*; *Ratnam et al., 2017*), providing new support to the long-standing proposal that the visual system uses oculo-motor induced luminance fluctuations for encoding spatial information in a temporal format (see *Rucci and Victor, 2015b* and *Rucci et al., 2018* for reviews). Building upon this previous work, here, we investigate whether this temporal encoding strategy, coupled with the known response characteristics of retinal neurons, accounts for the most fundamental properties of human spatial sensitivity.

In addition to the properties described above, it is well established that contrast sensitivity is affected by temporal modulations in the stimulus. Although the CSF exhibits a strong attenuation at low spatial frequencies when tested with stationary gratings, the shape of this function changes when gratings are modulated in time, transitioning from band-pass to low-pass as the temporal frequency of the stimulus increases (*Robson, 1966*). Furthermore, although strongly attenuated, sensitivity also tends to shift to higher spatial frequencies when retinal image motion is strongly reduced, as in experiments of retinal stabilization (*Kelly, 1979*). In both these conditions, the temporal modulations impinging onto retinal receptors differ drastically from those generated by normal eye drift over stationary gratings.

Does a temporal strategy of spatial encoding reconcile neurophysiological and behavioral measurements of contrast sensitivity? And does this strategy explain the differences in the CSF measured in various experimental conditions? More broadly, does the oculomotor-driven dynamics of retinal ganglion cells provide a unified account of human spatial sensitivity? Answers to these questions are not only critical for advancing our comprehension of the mechanisms of visual encoding but also for understanding the consequences of abnormal retinal image motion and their clinical implications. In the following, we use neuronal models to quantitatively examine the impact of eye drift on neural activity and compare the responses of retinal ganglion cells to the CSF of primates.

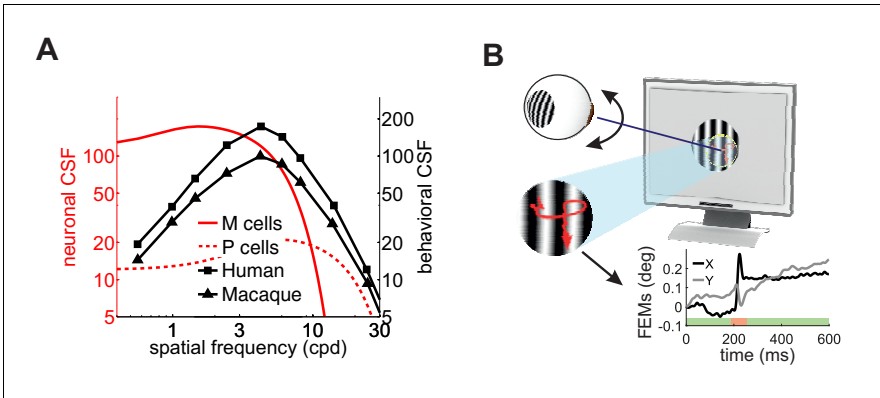

**Figure 1.** Contrast sensitivity and fixational eye movements. (A) Behavioral and neurophysiological measurements of contrast sensitivity. The contrast sensitivity functions (CSF) of humans and macaques (black curves; *De Valois et al., 1974*) are compared to the receptive fields profiles of magno- (M) and parvo-cellular (P) retinal ganglion cells (red curves; *Croner and Kaplan, 1995*). (B) Fixational eye movements (FEMs; red curve in magnified inset and black and gray traces in Cartesian graph), which include small saccades (microsaccades; red-shaded interval) and fixational drift (green), continually displace the stimulus on the retina.
DOI: https://doi.org/10.7554/eLife.40924.002

The following figure supplement is available for figure 1:

**Figure supplement 1.** Parametric analysis of the spatial sensitivity of magno- and parvo-cellular neurons.
DOI: https://doi.org/10.7554/eLife.40924.003

## Results

*Figure 1A* compares the mean receptive fields of ganglion cells in the primate retina, as estimated by *Croner and Kaplan (1995)*, with the contrast sensitivity of alert and behaving macaques (*De Valois et al., 1974*). The two sets of data deviate considerably, especially at low spatial frequencies. In this range, unlike the CSF, neural sensitivity is not strongly attenuated, a trend reported by multiple neurophysiological studies (e.g., *Kaplan and Shapley, 1982*; *Hicks et al., 1983*; *Derrington and Lennie, 1984*). This deviation is not simply the outcome of incorrectly extrapolating receptive-field measurements, as neural responses have been directly measured at very low spatial frequencies (down to 0.07 cpd in *Croner and Kaplan, 1995*; *Figure 1A*).

While a difference-of-Gaussians model can yield reduced responses at low spatial frequencies, attenuation similar to that observed in the CSF can only be achieved at the expense of highly unrealistic model parameters. As shown in *Figure 1—figure supplement 1A–B*, for both M and P cells, matching the physiological CSF requires a surround strength that is more than twice the value found in physiological measurements, a condition that gives an almost perfect balance between excitation and inhibition. Even small deviations from this balance lead to marked departures from the CSF (*Figure 1—figure supplement 1C–D*). Thus, contrary to previous proposals, the spatial sensitivity of retinal ganglion cells appears to be quantitatively incompatible with the characteristics of the CSF. A greater attenuation of neural sensitivity is required at low spatial frequencies to counterbalance the large power of natural scenes in this range.

The response of a neuron, however, does depend not only on the cell's spatial preference but also on its temporal sensitivity. Temporal transients are always present in the input signals to the retina during behavioral measurements of contrast sensitivity. Experimenters often take great care to minimize these transients, for example by slowly ramping up the stimulus at the beginning and down at the end of a trial and by enforcing fixation to prevent visual changes caused by saccadic eye movements (*Figure 2A*). Yet, despite these precautions, fixational eye movements are always present and modulate the visual flow impinging on the retina even when the stimulus does not change on the monitor. Could sensitivity to these oculomotor fluctuations reconcile neurophysiological and behavioral measurements of spatial sensitivity?

To investigate this question, we recorded eye movements in human observers, as they carried out a grating detection task at threshold and exposed spatiotemporal filters approximating the receptive fields of retinal ganglion cells to the luminance signals experienced by the retina in each

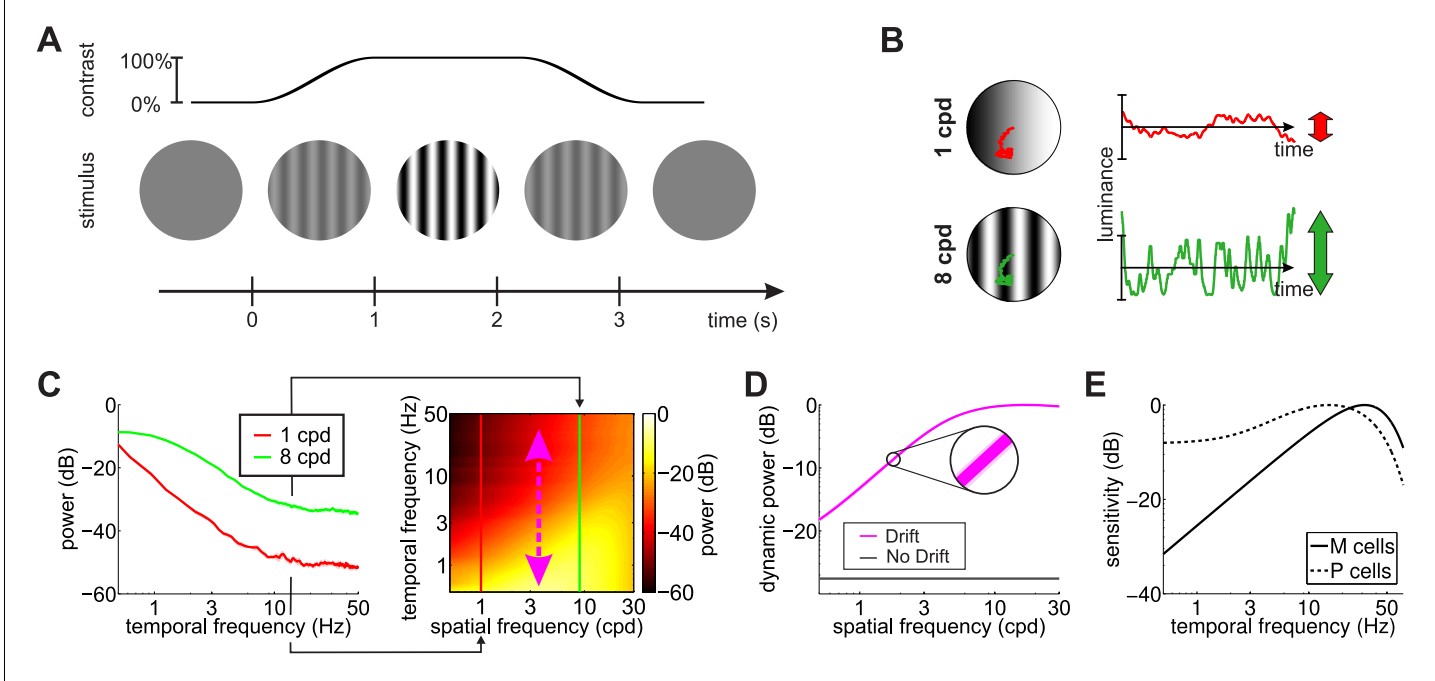

**Figure 2.** Input transients during measurement of contrast sensitivity. (**A**) Temporal modulations in the stimulus. Measurements of contrast sensitivity often change gradually the contrast of the stimulus during the course of the trial. In this case, the stimulus is a static grating. (**B**) Fixational jitter modulates input signals in a way that depends on the spatial frequency of the stimulus. The same amount of fixational drift yields larger temporal fluctuations with gratings at higher spatial frequencies (vertical arrows). (**C**) Input power with gratings at 1 and 8 cycles/deg (left panel). Higher spatial frequencies lead to broader temporal distributions (right panel). (**D**) Total power available at non-zero temporal frequencies with and without fixational drift. In this latter case, temporal modulations are only caused by the temporal contrast envelope of stimulus presentation. Shaded regions represents one standard deviation (see inset). Data represent averages over $N = 5$ observers. (**E**) Temporal sensitivities of modeled retinal ganglion cells. Model parameters are reported in *Tables 1* and *2* in Materials and methods.

DOI: https://doi.org/10.7554/eLife.40924.004

The following figure supplement is available for figure 2:

**Figure supplement 1.** Total power available in the retinal input at 0 Hz (static) with and without fixational drift.

DOI: https://doi.org/10.7554/eLife.40924.005

individual trial. *Figure 2B* shows the temporal modulations impinging onto retinal neurons during a typical measurement of contrast sensitivity. In the absence of any transient, the power of a stationary visual stimulus would be confined to the DC (0 Hz) temporal frequency axis. In practice, however, both eye drift and the turning of the stimulus on and off on the display introduce temporal modulations. These modulations effectively redistribute part of the stimulus DC power to nonzero temporal frequencies, that is they transform static power (the original power at 0 Hz) into dynamic power (power at non-zero temporal frequencies).

As shown in *Figure 2C–D*, because of the characteristics of ocular drift, the resulting dynamic power increases with spatial frequency, up to approximately 30 cpd (magenta line in *Figure 2D*), which, interestingly, roughly corresponds to the frequency limit given by the spatial resolution of photoreceptors in the fovea. In contrast, unlike drift, contrast modulations due to the onset/offset of the stimulus on the display cause power redistributions that do not depend on the spatial frequency of the stimulus (black line in *Figure 2D*). It is important to keep in mind that eye movements do not generate new power in the retinal input. They only redistribute the original DC power of the stimulus, so that a complementary frequency-dependent attenuation of power occurs along the 0 Hz axis (*Figure 2—figure supplement 1*).

Both eye drift and contrast changes yield temporal modulations that are well within the range of temporal sensitivity of retinal ganglion cells (cfg. *Figure 2C and E*). However, in simulations that replicated the standard conditions of contrast sensitivity measurements, drift modulations predominated. Since drift modulations convey little power at low spatial frequencies, the responses of standard ganglion cells were attenuated in this frequency range (*Figure 3B–C*). This happened for

both M and P cells, despite the well-known differences in their spatio-temporal sensitivity. As a consequence of this effect, a simple linear combination of the resulting M and P responses accurately predicted human contrast sensitivity with stationary stimuli over the entire range of relevant spatial frequencies (solid line in *Figure 3A*).

In contrast, in the absence of eye movements, when the only temporal modulations were those given by the onset/offset of the stimulus on the monitor, the CSF predicted by the same linear combination of neural responses exhibited a low-pass behavior that deviated considerably from human contrast sensitivity, especially at low spatial frequencies (dashed lines in *Figure 3*). In fact, no linear combination of modeled responses could approximate the CSF in this condition. This happened because, unlike the luminance modulations resulting from ocular drift, the amplitude of the contrast modulations of the stimulus on the display does not depend on the spatial frequency of the stimulus (black line in *Figure 2D*). Thus, without taking ocular drift into account, neuronal models exhibit a higher level of response at low spatial frequencies, as dictated by the spatial sensitivity of their kernels — and this strongly deviates from the CSF (*Figure 1A*).

In sum, standard models of the responses of M and P RGCs well predict the shape of the human CSF as measured with stationary gratings, but only when one considers sensitivity to the temporal modulations caused on the retina by fixational drift.

Contrast sensitivity is a function not only of the spatial frequency of the stimulus but also of its temporal frequency. Measurements with gratings modulated in time have long shown that the CSF in humans is not space-time separable: the way contrast sensitivity varies with spatial frequency depends on the temporal frequency of the modulation (*Robson, 1966*). As the temporal frequency increases, the CSF changes its shape, transitioning from band-pass to low-pass (*Figure 4A*).

To investigate whether our model also accounts for this change in shape, we repeated our simulations using gratings modulated at various temporal frequencies. The same linear combination of the responses of M and P cells as in *Figure 3* continued to closely match human performance when the stimulus was temporally modulated on the display, and the predicted CSF replicated the low-pass to band-pass transition observed in primates, as the frequency of the modulation increased (*Figure 4B*).

This change in shape was the consequence of the different amount of dynamic power that the combination of fixational drift and temporal modulations of the stimulus delivered within the range of neuronal sensitivity. Since we assume that there is no sensitivity to unchanging stimuli, the DC power does not contribute to cells' responses. However, flickering a grating has the effect of shifting the 0 Hz power of the grating to the temporal frequency of the modulation (*Figure 4C*). As a consequence, as the frequency of the modulation increased, this DC power was progressively moved into the sensitivity range of modeled neurons. At low temporal modulating frequencies (e.g. 1 Hz or below), only a small fraction of this power was within the region of neuronal sensitivity, and the temporal redistribution resulting from eye drift continued to exert a strong influence, forcing the CSF to maintain its band-pass shape. However, at higher temporal frequencies (e.g. 6 Hz and higher), the power restricted to the 0 Hz axis in the absence of stimulus' modulations now became fully available within the cells' peak sensitivity region. Since this static power is predominantly at low spatial frequencies (*Figure 2—figure supplement 1*), it caused a transition from band-pass to low-pass behavior in the responses of simulated M and P neurons, as well as in the shape of the CSF. Estimates of the CSF at intermediate frequencies between 0 Hz and 6 Hz (*Figure 4—figure supplement 1*) suggest that this transition occurs around 3 Hz, which is in agreement with psychophysical results (*Bowker and Tulunay-Keesey, 1983*).

In sum, our model attributes the space-time inseparability of the CSF to the structure of the temporal modulations delivered within the range of sensitivity of retinal ganglion cells. Modulations resulting from eye drift yield a band-pass CSF, whereas sinusoidally modulated gratings yield a low-pass CSF. The interplay between these two components of the retinal input explains not only contrast sensitivity with stationary gratings, but also the band-pass to low-pass transition that occurs with temporally modulated gratings. Notably, it correctly predicts the temporal frequency range at which this transition takes place. Our results, thus, suggest a functional link between the physiological instability of visual fixation and the characteristics of the CSF.

A natural question then emerges: how is contrast sensitivity affected by elimination of the luminance modulations caused by ocular drift? Ideally, in the complete absence of eye movements, neural responses in our model would only be driven by the modulations present in the external stimulus.

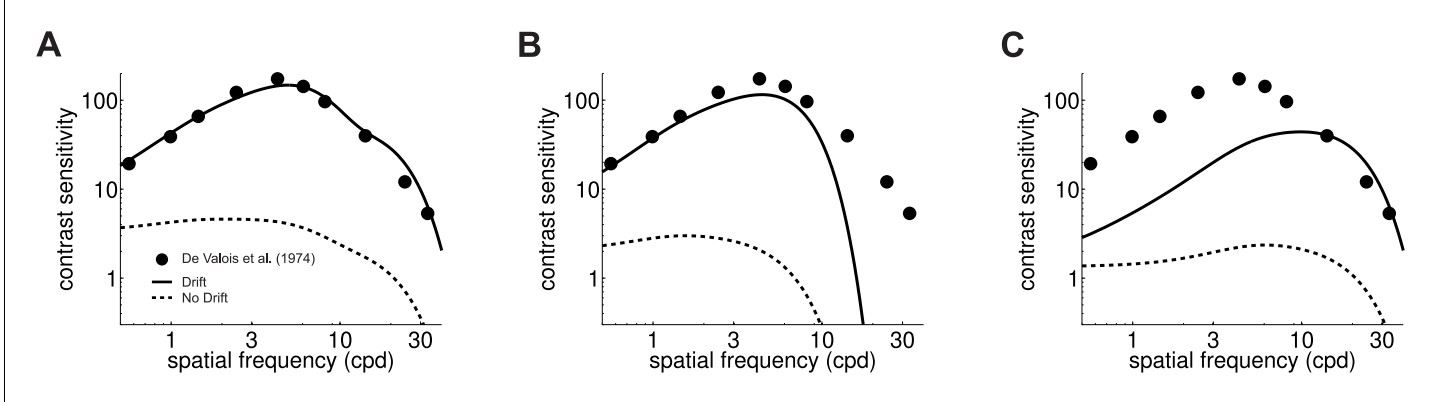

**Figure 3.** Influence of fixational drift on contrast sensitivity. Predicted CSFs in the presence (Drift; solid line) and absence (No Drift; dashed line) of eye movements. Stimuli were stationary gratings. (**A**) A linear combination of the responses of M and P cells closely matches classical measurements (circles; data from *De Valois et al., 1974*) only when eye drift occurs. (**B–C**) CSFs predicted separately from the responses of M (panel B) and P (panel C) cells.

DOI: https://doi.org/10.7554/eLife.40924.008

The following source data is available for figure 3:

**Source data 1.** Source data for *Figures 3* and *4* loaded by *Source code 1* and *2*.
DOI: https://doi.org/10.7554/eLife.40924.009

Under such conditions, the model predicts that sensitivity to a stationary grating would be greatly attenuated and the CSF would shift toward a low-pass shape, as it would lack the frequency-dependent amplification operated by ocular drift.

In real experiments, however, elimination of oculomotor-induced luminance modulations is impossible. Retinal stabilization — a laboratory procedure that attempts to immobilize an image on

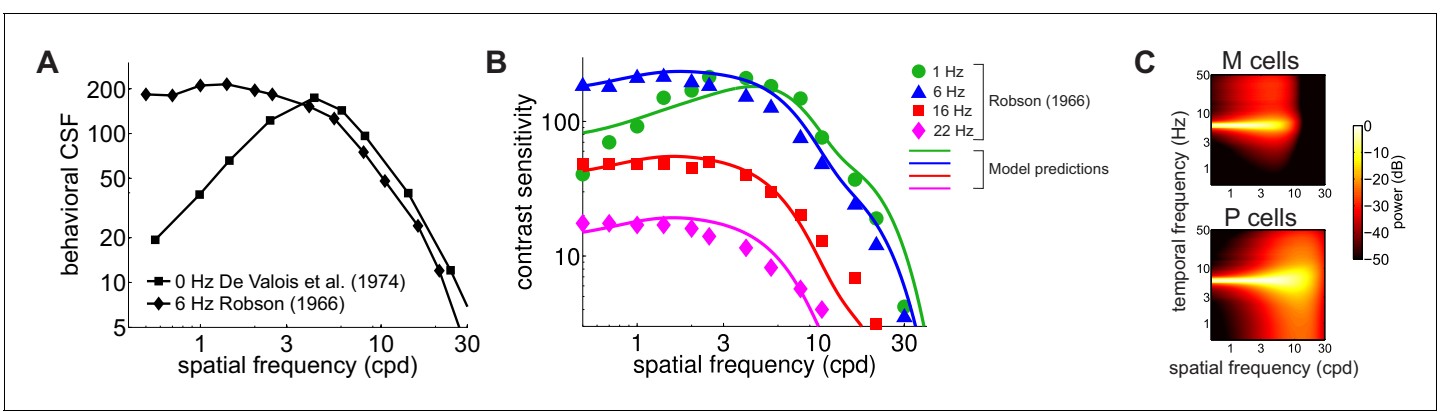

**Figure 4.** Contributions of fixational drift to contrast sensitivity with temporally modulated gratings. (**A**) Human CSFs measured with static (0 Hz; data from *De Valois et al., 1974* and sinusoidally modulated (6 Hz; data from *Robson, 1966*) gratings. (**B**) Contrast sensitivity functions predicted by our model in the presence of temporally modulated gratings are compared with measurements from *Robson (1966)*. See *Figure 4—figure supplement 3* for the separate contributions of M and P cells. (**C**) Power spectra of the response of modeled retinal ganglion cells during viewing of gratings temporally modulated at 6 Hz. Each point in the map represents the amount of power at a given temporal frequency resulting from translating the modeled receptive fields over a grating at the corresponding spatial frequency following the recorded eye drift trajectories.

DOI: https://doi.org/10.7554/eLife.40924.010

The following figure supplements are available for figure 4:

**Figure supplement 1.** Predicted CSF during normal viewing of temporally modulated gratings.
DOI: https://doi.org/10.7554/eLife.40924.011
**Figure supplement 2.** Robustness of the model to the specific implementation of the reduction in sensitivity at low temporal frequencies.
DOI: https://doi.org/10.7554/eLife.40924.012
**Figure supplement 3.** CSF predicted separately from the responses of M and P cells.
DOI: https://doi.org/10.7554/eLife.40924.013

the retina (*Riggs et al., 1953*; *Yarbus, 1957*) — is always affected by noise in the oculomotor recordings as well as imperfections in gaze-contingent display control, which leave some residual motion on the retina. Under these conditions, contrast sensitivity has indeed been found to be attenuated, but it maintains its band-pass shape and peaks at higher spatial frequencies (*Kelly, 1979*).

To examine whether sensitivity to temporal transients accounts for the changes in the CSF measured under retinal stabilization, we exposed modeled neurons to reconstructions of the visual input signals experienced in these experiments. Previous studies have established that a Brownian model well captures the characteristics of retinal image motion during fixation (*Kuang et al., 2012*; *Poletti et al., 2015*). Building on this previous finding, we modeled the residual motion of the retinal image in stabilization experiments as a Brownian process, but with greatly reduced diffusion coefficients relative to that present during normal, unstabilized fixation.

*Figure 5A* shows how the spatial frequency content of the luminance fluctuations experienced by retinal receptors (the power available at nonzero temporal frequencies) varies with the scale of the Brownian motion process (i.e. its diffusion coefficient, $D$). Changing the amount of retinal image motion has interesting repercussions on the characteristics of temporal modulations. As expected, a smaller diffusion constant delivers less dynamic power to the retina within the range of neural sensitivity, a direct consequence of the fact that luminance modulations are now smaller. However, a smaller $D$ also has the effect of shifting the range of amplification to higher spatial frequencies by a factor of $\sqrt{D}$. This happens because reducing the scale of retinal image motion is functionally equivalent to spatially stretching the stimulus, which translates, in the Fourier domain, to a compression of the axis of spatial frequencies that moves the amplification range toward higher spatial frequencies.

These effects in the spectral distributions of the retinal flow well match the changes in contrast sensitivity observed in retinal stabilization experiments. *Figure 5B* compares classical retinal stabilization data from *Kelly (1979)* to the sensitivity predicted by our model when the diffusion constant of the retinal image motion was attenuated by a factor of 125, which corresponds to shrinking the spatial scale of eye movements by approximately one order of magnitude. Model predictions closely followed psychophysical measurements: a reduction in the amount of retinal image motion attenuated contrast sensitivity while maintaining its band-pass shape and shifted its peak sensitivity to higher spatial frequencies from 4 Hz to 5.5 Hz (*Figure 5B*).

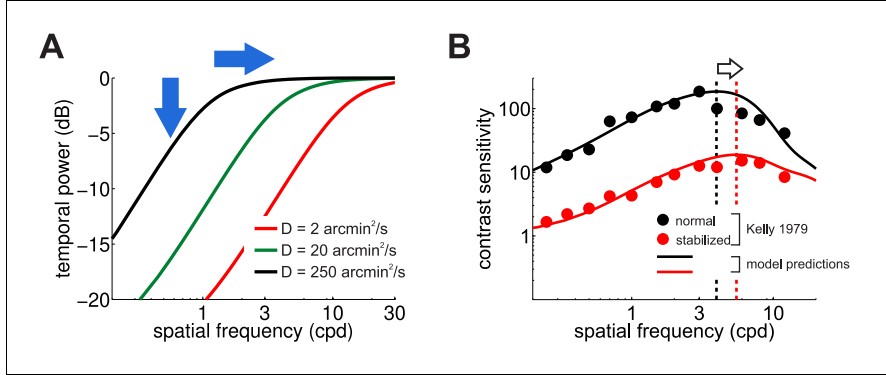

**Figure 5.** Consequences of retinal stabilization. (**A**) Spatial spectral density of the luminance modulations resulting from a Brownian model of retinal image motion with different diffusion constants. Lowering $D$ both attenuates the power available at each spatial frequency (vertical arrow) and shifts the distribution to higher spatial frequencies (horizontal arrow). (**B**) Predicted contrast sensitivity under retinal stabilization. Sensitivity is reduced and shifted to higher spatial frequencies. Dashed vertical lines mark the maxima of the two curves (color coded according to their $D$ in panel A). Results quantitatively match classical experimental data from *Kelly (1979)*. CSFs predicted separately from the responses of M and P neurons are shown in *Figure 5—figure supplement 1*.
DOI: https://doi.org/10.7554/eLife.40924.014

The following source data and figure supplement are available for figure 5:

**Source data 1.** Source data loaded by *Source code 3*
DOI: https://doi.org/10.7554/eLife.40924.015

**Figure supplement 1.** CSF predicted separately from the responses of M and P cells.
DOI: https://doi.org/10.7554/eLife.40924.016

These data show that consideration of the luminance modulations resulting from the motion of the stimulus on the retina accounts not only for behavioral sensitivity measurements performed in the presence of normal eye movements, but also for measurements made under conditions of retinal stabilization, when retinal image motion is greatly reduced.

## Discussion

Contrast sensitivity is a fundamental descriptor of visual functions. In many species, including humans, sensitivity strongly depends on the spatial and temporal frequency of the stimulus. Here, we show that a temporal scheme of spatial encoding, a scheme in which spatial vision is driven by temporal changes, predicts such dependencies when the temporal modulations introduced by incessant eye movements are taken into account. In contrast, when these consequences of fixational drift are ignored, the known response characteristics of retinal ganglion cells fail to account for human CSF. As described below, these results are highly robust, bear multiple consequences, and lead to important predictions.

An important consequence of our results regards the strategies by which the visual system encodes spatial information. Existing theories of visual processing have attributed the shape of the CSF to the characteristics of early visual processing. In an influential study (*Atick and Redlich, 1992*) found that the theoretical filter that optimally decorrelates natural images closely matches the CSF. Since decorrelated responses enable compact neural representations, these authors assumed that the CSF reflects the average spatial selectivity of ganglion cells in the retina. However, experimental measurements have long shown that the response selectivity of RGCs differs considerably from the CSF, particularly at low spatial frequencies, where decorrelation would be most beneficial (*Hicks et al., 1983*; *Kaplan and Shapley, 1982*; *Derrington and Lennie, 1984*; *Croner and Kaplan, 1995*). As expected from this deviation, broad spatial correlations in RGCs responses have been found in preparations in which natural images are displayed in the absence of eye movements (*Puchalla et al., 2005*; *Segal et al., 2015*). These findings are consistent with our model: when the transients in stimulus presentation override the consequences of eye drift, spatial sensitivity follows the spatial kernels of modeled receptive fields. For this reason, responses to low spatial frequencies are enhanced relative to the level that would be needed for decorrelating activity.

The same principle also provides an explanation for the band-pass to low-pass transition of the CSF as the temporal frequency of the stimulus increases. This transition is the consequence of the spectral characteristics of the signals that the combination of fixational drift and stimulus transients delivers within the range of neuronal temporal sensitivity. With stationary gratings, temporal modulations in the retinal input are heavily influenced by ocular drift, which enhances high spatial frequencies imposing a band-pass sensitivity (*Figure 3*). With temporally modulated gratings, neuronal responses are also affected by the contrast modulation imposed to the stimulus on the display. Above a frequency of a few Hz, the impact of external modulations outweighs the effects of eye movements, removes the space-time inseparability in cell responses caused by ocular drift, and enhances again sensitivity to low spatial frequencies (*Figure 4B*).

Rather than attributing spatial sensitivity solely to the spatial selectivity of RGCs, our analysis shows that the CSF is shaped by the joint *spatial* and *temporal* characteristics of retinal responses and how they interact with oculomotor transients. It predicts the complex way contrast sensitivity varies with the spatial and temporal frequency of the stimulus by a linear combination of the space-time separable functions of P and M channels. While our study cannot exclude that other mechanisms, at various stages of visual processing, may also play a role in shaping the CSF (e.g. the number of neurons in different frequency channels), it suggests that these other contributions are minimal. Consideration of RGCs temporal sensitivity provides a parsimonious unifying framework for a wide range of experimental measurements of the CSF with only a minimal set of assumptions.

In our model, we assumed that retinal ganglion cells possess negligible sensitivity below the frequencies at which sensitivity can practically be measured (~0.2–0.3 Hz). This hypothesis may appear to conflict with the neurophysiological data reported in the low temporal frequency range by several studies. However, in both neurophysiological and psychophysical experiments, measuring sensitivity in this range is challenging because it requires trials with long durations, consideration of the visual stimuli present before and after each trial, and estimation of long impulse responses. Typically, the transfer functions reported at low temporal frequencies are extrapolations outside of the range of

measured values based on models that were not designed for this purpose (e.g. the linear cascade model (*Victor, 1987*) in *Benardete and Kaplan, 1997b*; *Benardete and Kaplan, 1997a*; a difference of exponential in *Derrington and Lennie, 1984*, etc.). These extrapolations must be interpreted with great caution, as they merely reflect untested model assumptions.

The few studies that specifically examined retinal ganglion cells' responses at low temporal frequencies found a decline in sensitivity up to the limit that they could measure (*Victor, 1987*; *Purpura et al., 1990*). These studies suggest that the response attenuation takes the form of an approximately linear decrease in log-log scale. Such behavior is expected from theoretical considerations based on the characteristics of adaptation (*Thorson and Biederman-Thorson, 1974*), considerations that appear to apply to the responses of cones in the retina of the macaque (*Boynton and Whitten, 1970*) and therefore will limit the low-frequency behavior of retinal ganglion cells. Furthermore, temporal signals at frequencies below ~0.3 Hz, even if present, are not likely to be useful to an observer in a psychophysical experiment, as they will contain noise power due to visual stimulation on previous trials and during the intertrial interval (such as eye-blinks and glances around the lab). Our results are robust to the specifics of how this low-frequency attenuation in sensitivity was implemented. The curves presented in *Figures 3*, *4* and *5* were obtained by simply discarding responses below 0.6 Hz. Results were, however, virtually identical when we used different frequency thresholds (*Figure 4—figure supplement 2A*), or when we modeled sensitivity as a power law function in the low-frequency range, as in *Purpura et al. (1990)* and *Thorson and Biederman-Thorson (1974)* (*Figure 4—figure supplement 2B*).

We specifically focused on fixational drift both because of its ubiquitous presence and its known influence on fine pattern vision (*Ratliff and Riggs, 1950*; *Ditchburn, 1955*; *Steinman et al., 1973*; *Rucci et al., 2007*; *Ratnam et al., 2017*). Other types of eye movements, like saccades and microsaccades, tend to be suppressed during measurements of contrast sensitivity (*Mostofi et al., 2016*) and were not considered in this study. The transients from these movements, however, differ in their spectra from those from eye drift, as they provide equal temporal power across a broad range of spatial frequencies. Thus, during normal viewing, the visual system could benefit from different types of modulations. In keeping with this idea, it has been argued that the stereotypical alternation of oculomotor transients resulting from the natural saccade/drift cycle contributes to a coarse-to-fine processing dynamics at each visual fixation (*Boi et al., 2017*).

It is worth emphasizing that our results are very robust and do not depend on fitting model parameters. With regard to oculomotor activity, we did not model eye movements, but used real traces recorded from human subjects during measurements of contrast sensitivity. With regard to neuronal properties, we implemented standard M and P filters obtained from the neurophysiological literature and frequently adopted by modeling studies (*Croner and Kaplan, 1995*; *Benardete and Kaplan, 1997a*; *Benardete and Kaplan, 1999*). We chose to estimate the CSF by linearly combining M and P responses in fixed ratio, because this was the simplest model. But we note that other ways of combining M and P signals will yield very similar conclusions, since the space-time inseparability originate from the visual input rather than the neuronal models. Our two parameters (the global gain at a given temporal frequency and the ratio of M-P contributions, see *Equation 7* in the Materials and methods section) were merely used to quantitatively align the modeled CSF with the experimental data. They have no role in explaining the shape of the CSF and its band- to low-pass transition.

In addition to providing a comprehensive explanation of the CSF, our study makes important predictions at different levels. At the neural level, our results predict that the response selectivity of RGCs will change when measured in the presence and absence of the fixational motion of the retinal image. Neurophysiological studies already suggest that fixational eye movements are an important component of visual encoding (*Gur et al., 1997*; *Leopold and Logothetis, 1998*; *Martinez-Conde et al., 2000*; *Olveczky et al., 2003*; *Kagan et al., 2008*; *Meirovithz et al., 2012*; *McFarland et al., 2016*). Eye jitter has been found to reduce redundancy in the responses of retinal neurons (*Segal et al., 2015*) and to synchronize them, enhancing visual features (*Greschner et al., 2002*) even beyond the physiological limitations imposed by photoreceptors spacing (*Juusola et al., 2016*). Furthermore, retinal ganglion cells have been found that may distinguish between the global motion given by fixational eye movements and the local motion of objects (*Olveczky et al., 2003*). Yet, retinal responses are traditionally measured with the eyes immobilized, a condition in which RGCs tend to exhibit relatively strong responses at low spatial frequencies (*Croner and Kaplan,*

*1995*). Our model predicts that the spatial frequency amplification produced by fixational drift in the retinal input (*Figure 2D*) will enhance neuronal sensitivity to higher spatial frequencies and will reduce sensitivity to low spatial frequencies. As a consequence, RGCs' spatial sensitivity should exhibit a more pronounced band-pass behavior and its peak should shift toward higher frequencies. This prediction is difficult to test in vivo, because of the need to completely stabilize the retinal input, but it can be thoroughly investigated in vitro, where the motion of the retinal image is under full experimental control.

At the perceptual level, an interesting observation comes from the changes in the frequency content of the retinal input shown in *Figure 5A*. The amplitude of fixational instability regulates the power available in different spatial frequency bands. Specifically, the smaller the amount of retinal image motion, the more the range of amplification shifts to higher spatial frequencies. The visual system could, in principle, exploit this relationship by dynamically matching the spatial scale of eye drift to the frequency content of the visual scene, or the frequency range that is task-relevant. Within a certain range, smaller drifts would optimize information accrual when foveating on regions rich in high spatial frequencies. This effect could not only be directly driven by the stimulus in a bottom-up fashion, but also be used to meet top-down demands in high-acuity tasks. Indeed, several studies support the idea that humans can control the amount of their ocular drift (*Steinman et al., 1973*; *Cherici et al., 2012*; *Poletti et al., 2015*). In the same vein, the relationship between fixational drift and the frequency content of the retinal input may also explain individual perceptual differences. Subjects with relatively smaller drifts are expected to perform better in tasks in which high spatial frequencies are critical. Studies that quantitatively relate the characteristics of fixational eye drift to visual perception are needed to investigate these predictions.

Furthermore, our model predicts that manipulating temporal modulations from eye drift will affect performance. We have shown that reducing the amount of the retinal jitter well matches the overall reduction in contrast sensitivity as well as the shift to higher spatial frequencies observed in experiments of retinal stabilization. In the other direction, enlarging fixational jitter increases the amount of power available at low spatial frequencies predicting an improvement in contrast sensitivity in this range. This prediction is consistent with the improvements in word and object recognition reported in patients with central visual loss, when images or text are jittered or scrolled (*Watson et al., 2012*; *Harvey and Walker, 2014*; *Gustafsson and Inde, 2004*). The spatial frequency band of retinal ganglion cells decreases with eccentricity and enlarging retinal image motion has the effect of bringing more power in their range of sensitivity.

Our study also has clinical implications, as it predicts that disturbances in fixational oculomotor control will affect visual sensitivity. Oculomotor anomalies and impaired sensitivity co-occur in a variety of disorders, including conditions as diverse as dyslexia (*Stein and Fowler, 1981*; *Stein and Fowler, 1993*) and schizophrenia (*Dowiasch et al., 2016*; *Egaña et al., 2013*). Patients with these conditions exhibit similar visual deficits including reduced sensitivity (*Lovegrove et al., 1980a*; *Lovegrove et al., 1980b*; *Slaghuis, 1998*), low-level visual impairments (*Eden et al., 1996*; *Li, 2002*; *Butler et al., 2001*; *Kim et al., 2006*) and reading disabilities (*Revheim et al., 2006*) possibly caused by the disturbances in low-level vision (*Revheim et al., 2006*; *Lovegrove et al., 1980a*). Our results suggest a potential link between fine-scale eye movements and these visual deficits, which has not yet been investigated and which may inspire novel therapeutic approaches.

## Materials and methods

### Data collection and analysis

To examine the influences of eye movements on visual sensitivity, neuronal models were exposed to reconstructions of the input signals typically experienced by observers in experiments of contrast sensitivity. To this end, we used oculomotor traces recorded in measurements of contrast sensitivity to move the stimuli presented as input to the models. Methods for the collection and analysis of eye movements data, as well as perceptual results have already been described in previous publications and are only briefly summarized here (see *Mostofi et al., 2016* and *Boi et al., 2017*). This section focuses on the methods that are novel to this study.

## Subjects

Eye movements were recorded from five observers (all females, age range 21–31). To optimize the precision of the recordings, only subjects with normal, uncorrected vision took part in the study. Informed consent was obtained from all participants following the procedures approved by the Boston University Charles River Campus Institutional Review Board (protocol number 1062E).

## Apparatus

Stimuli were displayed on a gamma-corrected fast-phosphor CRT monitor (Iyama HM204DT) in a dimly-illuminated room. They were observed monocularly with the left eye patched, while movements of the right eye were recorded by means of a Dual Purkinje Image eyetracker (Fourward Technology) and sampled at 1 KHz. This system has a resolution – measured by means of an artificial eye – of approximately 1'(*Crane and Steele, 1985*; *Ko et al., 2016*). A dental imprint bite bar and a head-rest prevented head movements. Stimuli were rendered by means of EyeRIS, a custom system that enables precise synchronization between oculomotor events and the refresh of the image on the monitor (*Santini et al., 2007*).

## Stimuli and procedure

As in typical psychophysical CSF measurements, we used a standard grating-detection paradigm (see *Mostofi et al., 2016* for the behavioral data). In a forced-choice procedure, observers detected 2D Gabor patterns oriented at $\pm45°$. Their contrast varied across trials following PEST (*Taylor and Creelman, 1967*). The frequency and standard deviation of the Gabor were 10 cycles/deg and 2.25° respectively. Stimuli were displayed over a uniform field with luminance of 21 $cd/m^2$. Oculomotor traces were segmented in complementary periods of drift and saccades based on a speed threshold of $2^o$/s (*Mostofi et al., 2016*). Only oculomotor traces collected around threshold levels of sensitivity and that contained no saccades, microsaccades or blinks were used in this study.

Modeled neurons were exposed to the same retinal input experienced by human participants, identically replicated at all spatial frequencies. Gratings were presented for 3.2 s. They were smoothly ramped up and down in contrast at the beginning and end of the trial by means of the modulating function $M(t)$ and also modulated in time at frequency $\omega_t$ ($\omega_t$ = 0, 1, 6, 16, or 22 Hz). The reconstructed retinal input was thus given by:

$$I(\boldsymbol{x}, t|f_s, \omega_t, \alpha_s, \phi_s) = sin(2\pi \boldsymbol{f_s} \cdot (\boldsymbol{x} - \xi(t))^T + \phi_s) \cdot sin(2\pi\omega_t t) \cdot M(t) \qquad (1)$$

where $\xi(t) = [\xi_x(t), \xi_y(t)]$ represents eye movements and $\boldsymbol{f_s} = [f_s cos(\alpha_s), f_s sin(\alpha_s)]$ the stimulus frequency (0.1–60 cycles/deg). The orientation $\alpha_s$ and the phase $\phi_s$ uniformly spanned the range $[0 \ 2\pi]$.

## Neural models

The mean instantaneous rate of retinal ganglion cells (RGCs) were simulated by means of standard space-time separable linear filters with transfer function:

$$RF(\boldsymbol{f}, \omega) = K(\boldsymbol{f}) \cdot H(\omega) \qquad (2)$$

where $\boldsymbol{f}$ and $\omega$ indicate spatial and temporal frequencies respectively. The spatial kernel $K(\boldsymbol{f})$ was modeled as in *Croner and Kaplan (1995)* with a standard difference of Gaussians:

$$K(\boldsymbol{f}) = C(K_c \pi r_c^2 e^{-\pi r_c |\gamma f|^2} - K_s \pi r_s^2 e^{-\pi r_s |\gamma f|^2}) \qquad (3)$$

**Table 1.** Parameters used in *Equation 3* to model the spatial kernels of magno- (upper row) and parvo-cellular (bottom row) neurons.
Data are from *Croner and Kaplan (1995)*.

| | $r_c$ | $K_c$ | $r_s$ | $K_s$ |
|---|---|---|---|---|
| **M cells** | 0.10 | 148 | 0.72 | 1.1 |
| **P cells** | 0.03 | 353.2 | 0.18 | 4.4 |

DOI: https://doi.org/10.7554/eLife.40924.006

**Table 2.** Parameters used in *Equation 4* to model the temporal kernels of magno- (upper row) and parvo-cellular (bottom row) neurons.
Data are from *Benardete and Kaplan (1997a)*; *Benardete and Kaplan (1999)*.

| | $N$ | $A$ | $D$ | $H_s$ | $\tau_L$ | $\tau_S$ |
|---|---|---|---|---|---|---|
| **M cells** | 30 | 499.77 | 2 | 1 | 1.1 | 2.23 |
| **P cells** | 38 | 67.59 | 3.5 | 0.69 | 1.27 | 29.36 |

DOI: https://doi.org/10.7554/eLife.40924.007

with parameters adjusted based on neurophysiological recordings from macaques (*Table 1* in *Croner and Kaplan, 1995*). The scaling factor $\gamma$ was set to 0.5 to model the smaller receptive fields of the fovea following cortical magnification (Eq.8 in *Van Essen et al., 1984*).

The temporal sensitivity function $H(\omega)$ consisted of a series of low-pass filters and a high-pass stage as propose by *Victor (1987)*:

$$H(\omega) = Ae^{-i\rho 2\pi\omega D}\left(1 - \frac{H_s}{1 + i\rho 2\pi\omega\tau_S}\right)\left(\frac{1}{1 + i\rho 2\pi\omega\tau_L}\right)^N \tag{4}$$

Parameters were taken from neurophysiological studies that fitted this model to recorded neurons (M cells: median values in *Table 2* in *Benardete and Kaplan, 1999*; P cells: median values in *Table 2* in *Benardete and Kaplan, 1997a*). The scaling factor $\rho$ was set to 1/1.6 to include the effects of large stimuli on retinal responses (Figure 7B in *Alitto and Usrey, 2015*).

## Estimating contrast sensitivity

The main hypothesis of our study is that the visual system is insensitive to temporal stimulation at 0 Hz so that spatial sensitivity is entirely driven by temporal transients. For this reason, we estimated the predicted CSF on the basis of cell responses to input changes.

For each spatial frequency $f_s$ of the grating, we first estimated the space-time power spectrum of the retinal input $P_I(f, \omega)$ by averaging the square of the absolute value of the Fourier transform of *Equation 1* across trials, stimulus' orientations $\alpha_s$ and phases $\phi_s$. Since both $P_I(f, \omega)$ and the spatial kernels $K(f)$ possess circular symmetry in spatial frequency, we reduced the spatial dimensionality from 2D to 1D by radial averaging. We then computed the power spectrum of neuronal responses $O(f, \omega)$ by multiplying the space-time power spectrum of the retinal input $P_I(f, \omega)$ by the transfer functions of the cells' filters:

$$O_\zeta(f, \omega) = P_I(f, \omega) \cdot |RF_\zeta(f, \omega)|^2 \tag{5}$$

where $RF_\zeta(f, \omega)$, with $\zeta = M$ or $P$, represents the Fourier transform of M or P cells' receptive fields (*Equation 2*).

Finally, we evaluated the CSF at each spatial frequency $f$, by computing the square root of the integrated temporal power across all non-zero temporal frequencies:

$$CSF_\zeta(f) = \sqrt{\int_{o^+}^{\infty} O_\zeta(f, \omega)d\omega} \tag{6}$$

where $O_\zeta$ represents the power spectrum of M or P responses. The integral in *Equation 6* was computed numerically. To avoid artifacts from finite bandwidth, the first two temporal samples of the spectrum were discarded so that integral over temporal frequency started from $\omega = 0.63\,Hz$. However, virtually identical results were obtained when we used lower thresholds or when we modeled the low-frequency range of temporal sensitivity as a power law (*Figure 4—figure supplement 2*).

The predicted CSF was then estimated, for each condition, by a linear combination of the contrast sensitivities of the two types of neurons, $CSF_M(f)$ and $CSF_P(f)$ :

$$CSF_{est}(f) = A \cdot [\lambda CSF_M(f) + (1 - \lambda) \cdot CSF_P(f)] \tag{7}$$

where $\lambda$ ($\lambda = 0.57$ for all conditions) weighs the contributions of the M and P populations and $A$ is a global rescaling coefficient.

Note that the parameters $A$ and $\lambda$ were merely used to quantitatively align model predictions with classical data, but had no role in explaining our findings. That is, the emergence of a space-time inseparability in the CSF, was neither caused by the specific value of $\lambda$ (both M and P cells show this transition; *Figure 4—figure supplement 3*) nor by the global scaling factor $A$, which had no effect on the shape of the predicted CSF. We chose to linearly combine the contributions of M and P neurons because this was the simplest model. However, use of other models (e.g. the maximum of either population at each spatial frequency $f$) produced virtually the same results given the robustness of the underlying phenomenon.

The same procedure was used to estimate the CSF in the case of no eye movements and retinal stabilization (*Figures 3*, *4* and *5*). In the former condition (no eye movements), $\xi(t)$ was set to zero in *Equation 1*. In the latter condition (retinal stabilization), we modeled the retinal image motion by means of a 2D random walk process, but with reduced diffusion coefficient ($D = 2$ rather than the normal value $D = 250$). Brownian motion, with $D$ in the range 100–350, is known to be a good model for the normal retinal image motion when the head is not immobilized (*Aytekin et al., 2014*).

## Additional information

### Funding

| Funder | Grant reference number | Author |
| --- | --- | --- |
| National Eye Institute | EY018363 | Michele Rucci |
| National Science Foundation | BCS-1457238 | Michele Rucci |
| National Eye Institute | NEI 07977 | Jonathan D Victor |
| National Science Foundation | 1420212 | Michele Rucci |
| Harvard/MIT Joint Research Program | | Antonino Casile |

The funders had no role in study design, data collection and interpretation, or the decision to submit the work for publication.

### Author contributions

Antonino Casile, Conceptualization, Data curation, Software, Supervision, Investigation, Methodology, Writing—original draft, Writing—review and editing; Jonathan D Victor, Conceptualization, Methodology, Writing—review and editing; Michele Rucci, Conceptualization, Supervision, Methodology, Writing—review and editing

### Author ORCIDs

Antonino Casile  http://orcid.org/0000-0002-7824-9274
Jonathan D Victor  http://orcid.org/0000-0002-9293-0111
Michele Rucci  http://orcid.org/0000-0002-3066-1964

### Ethics

Human subjects: Informed consent was obtained from all participants following the procedures approved by the Boston University Charles River Campus Institutional Review Board (protocol number 1062E).

### Decision letter and Author response

Decision letter https://doi.org/10.7554/eLife.40924.022
Author response https://doi.org/10.7554/eLife.40924.023

## Additional files

### Supplementary files

• Source code 1. Source Matlab code to generate *Figures 3A–C* in the manuscript.

DOI: https://doi.org/10.7554/eLife.40924.017

• Source code 2. Source Matlab code to generate *Figures 4B* in the manuscript.
DOI: https://doi.org/10.7554/eLife.40924.018

• Source code 3. Source Matlab code to generate *Figures 5B* in the manuscript.
DOI: https://doi.org/10.7554/eLife.40924.019

• Transparent reporting form
DOI: https://doi.org/10.7554/eLife.40924.020

### Data availability

All data generated or analysed during this study are included in the manuscript and supporting files.

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
