## [Decision Letter]

[Editors’ note: a previous version of this study was rejected after peer review, but the authors submitted for reconsideration. The first decision letter after peer review is shown below.]

Thank you for submitting your work entitled "Changes in visual sensitivity reveal an active strategy for temporally encoding space" for consideration by *eLife*. Your article has been reviewed by three peer reviewers, one of whom is a member of our Board of Reviewing Editors, and the evaluation has been overseen by a Senior Editor. The reviewers have opted to remain anonymous.

Our decision has been reached after consultation between the reviewers. Based on these discussions and the individual reviews below, we regret to inform you that your work cannot be considered further, at least in present form. We would be willing to consider a revised paper if you thoroughly deal with the reviewers’ concerns.

All three reviewers agreed that the work was of broad interest. We also agreed that several issues needed to be strengthened before considering the paper further. These and several other issues are detailed in the individual reviews. The two most pressing issues follow:

1) Recent work from some of the same authors has deepened understanding of the relation between FEM and neural coding. The relation between the present work and this past work is not as clear as it needs to be in the Introduction, and as a result the advance that the present work represents is difficult to appreciate immediately.

2) At the heart of the paper is the discrepancy between physiological and psychophysical CSFs. But the ganglion cell examples in the first figure appear not to accurately reflect ganglion cell behavior since they do not include points at low frequencies (with instead a horizontal line from the lowest spatial frequency point plotted). Most ganglion cell measurements show some low frequency roll-off. A thorough justification of the properties used to represent measured ganglion cell CSFs is needed.

*Reviewer #1:*

This paper describes the interaction between eye movements and spatial coding. The paper starts by describing a discrepancy between spatial contrast sensitivity as measured behaviorally and as measured in responses of retinal or geniculate cells. It then proposes that this discrepancy originates because the neural recordings were made in the absence of eye movements, and that the discrepancy can be resolved when eye movements are taken into account. The case for the importance of eye movements, made here and in previous work from some of the same authors, is compelling. There are several issues with the present paper, however, that limit my enthusiasm:

1) M and P cell models. The literature contains numerous measurements of M and P cell spatial contrast sensitivity. From a scan of these papers (e.g. Derrington and Lennie, 1984; Kaplan and Shapley, 1982), measured spatial contrast sensitivity curves for both M and P cells fall, sometimes sharply, at low spatial frequencies. This fall is not evident in the responses depicted in Figure 1. Even the Cronor and Kaplan (1995) paper cited as the basis of Figure 1 shows a clear falloff at low spatial frequencies, and describes properties of the associated surround (related, the Benardete and Kaplan, 1997 reference in the text is about M sequences and temporal properties, so not immediately clear how it is relevant). I am not clear on why the red curves in Figure 1 should be extrapolated from their peak to a 0 spatial frequency asymptote; this extrapolation appears to be inconsistent with most experimental results. More generally, I think it is critical that the paper contains a thorough summary of the measured M and P properties across studies, and a clear justification of the M and P models used. Figure 1 presently undermines confidence in the central motivation for the paper.

2) Overlap with previous work. The content of Figures 1 and 2 is a review of previous work. Of particular importance, previous work from some of the same authors has established that eye movements decrease responses to low spatial frequencies in natural images. I am concerned, given that, about the intellectual advance of the present paper. Specifically, I think the paper would benefit from more experimental tests of the proposal. For example, are there measurements of spatial contrast sensitivity in awake behaving monkeys (i.e. with eye movements)? Recognizing that eye movements cannot be fully suppressed, can they still be manipulated to test the proposal (e.g. can they be increased)? Are there differences across eccentricity that could be exploited to provide additional tests?

*Reviewer #2:*

The contrast sensitivity function (CSF) is a fundamental characteristic of vision – it describes how the ability to see depends on the spatial frequency of the input. The CSF is believed to be due to processing limitations at the earliest stages of processing (i.e., at the retina). However, there is a long-known discrepancy between the CSF (measured by having subjects report) and the properties of retinal output signals (measured physiologically) – subjects are less sensitive to low spatial frequencies than you would expect from the signals recorded in the retina. This paper puts forward and tests a specific explanation for this discrepancy – subjects make small eye movements during fixation, and this introduces temporal modulations that shift the power spectrum of the visual input. The paper convincingly demonstrates that this explanation is adequate to explain the discrepancy, using high-quality measurements of eye movements in human subjects combined with computational modeling of retinal ganglion cell responses. Moreover, they extend this approach to show that reducing fixational eye movements would be expected to shift the CSF in ways that are consistent with previous studies using retinal stabilization.

Overall, this is an expertly conducted combination of computational modeling and eye movement recording that provides an elegant solution to a long-standing problem in vision science. My comments are aimed mainly at clarifying the presentation.

The paper refers to "fixational eye movements" or "FEMs" throughout, but what is meant by this is the slow drift component of fixational eye movements, and not microsaccades or oscillatory movements. A few points. First, the paper is reasonably clear on this point, but not completely and not everywhere. For example, the casual reader who only looks at the Abstract and skims the details (including the Results where the focus on drift is pointed out) might conclude that this applies to microsaccades, since they are the best-known component of FEMs. Is there a reason to avoid using the more specific term "ocular drift" rather than "FEMs" throughout, or at least, more often? Second, what is the impact of microsaccades? Would these also be expected to affect the CSF or is the scale of their temporal modulation too high to affect responses? This point should be clarified. Third, if ocular drift and microsaccades have distinguishable effects on CSF, then the authors should be especially careful with the use of the term "FEMs", especially in the Discussion, where it seems that the conclusions based on ocular drift appear to be generalized to all types of fixational eye movements.

In several places – starting with the title – the manuscript implies that there is an active strategy behind these effects of FEMs, rather than it being an incidental effect. To me, "active strategy" implies some reliable relationship between the circumstance and the subject's behavior. For example, the needle-threading task shows that microsaccades appear to be generated based on some sort of active strategy. For ocular drift, it is not clear that such a relationship has been established. Do you have some basis of this claim, or is it simply a speculation? If it is a speculation, the text should be written more conservatively to match.

Another plausible explanation for why the behavioral CSF does not match the properties of the retinal output is that some part of the rest of the visual system is responsible. Is there a basis for excluding this possibility?

*Reviewer #3:*

This paper explores how non-separable spatio-temporal frequency tuning of M and P cells combines with measured fixational eye movements to account for observed behaviorally measured visual contrast sensitivity. A minimal model agrees well with a range of experimental data (free viewing of gratings at different temporal frequencies as well as retinal stabilization experiments) and suggests that eye movements have been optimized for neuronal contrast sensitivity curves (this work) as well as the content of natural scenes (previous work from the same groups). Experimental tests are suggested, particularly for retina-in-a-dish style experiments, where movements typical of FEMs are rarely added to visual stimuli. Experiments are also suggested for human subjects and for some clinical applications.

1) The paper is clearly presented and the work is thorough and interesting, but there should be more emphasis in the Introduction on why it's so important to understand the discrepancies between neuronal and behavioral CSFs for stationary stimuli. This could be addressed with a bit of text that previews some of the Discussion points about the neural coding and potential clinical applications of these findings.

2) Some recap of existing retinal recordings with FEM-like perturbations (e.g. Greschner et al., 2002) should be added to the Discussion.

3) Figure 1A – no data points are apparent below about 2Hz for the P and M cells, right around where the neuronal and behavioral CSFs diverge. It would be nice to include some data points there, perhaps from other studies. If data points do exist at 0Hz, they should be made more apparent.

[Editors’ note: what now follows is the decision letter after the authors submitted for further consideration.]

Thank you for submitting your article "Contrast sensitivity reveals an oculomotor strategy for temporally encoding space" for consideration by *eLife*. Your article has been reviewed by three peer reviewers, one of whom is a member of our Board of Reviewing Editors, and the evaluation has been overseen by Timothy Behrens as the Senior Editor. The reviewers have opted to remain anonymous.

The reviewers have discussed the reviews with one another and the Reviewing Editor has drafted this decision to help you prepare a revised submission.

The revisions have improved the paper substantially. There is one outstanding point remaining about the ganglion cell model (see reviews for details). In consultation all of the reviewers agreed this was important to resolve.

*Reviewer #1:*

This is a revision of a paper about the relationship between fixational eye movements and spatial contrast sensitivity. The paper centers around the observation that the spatial contrast sensitivity function measured behaviorally in primates differs from that of retinal ganglion cells, particularly at low spatial frequencies. The paper argues that eye movements and the temporal sensitivity of ganglion cells account for this difference.

The paper has improved in revision. In particular, I found the Introduction more compelling, and the connection of the spatial ganglion cell models to past work is much less confusing with Figure 1 fixed. An issue that became clearer in the revised version of the paper is that the temporal model for the ganglion cells needs to be described more thoroughly. This and some smaller points follow.

The crux of the paper is how the dynamics of eye movements make nominally static spatial inputs dynamic, and this shifts static inputs into a temporal region where the ganglion cells are more responsive. The ganglion cell model in Equation 4 is taken straight from prior measurements, and hence is well supported in the literature (see suggestion below about including relevant parameters). However, this model does not predict a complete lack of response at a temporal frequency of 0 Hz, as assumed in the paper. There are several issues with how this is treated in the paper. First, the temporal model is central to the paper, and should not be relegated entirely to the Materials and methods (one suggestion would be to add a figure between the current Figure 2 and 3 showing the temporal response of the ganglion cell model). Second, is there experimental evidence that static components of the stimulus do not modulate the ganglion cell responses? This assumption should be clarified earlier (in reference to Figure 3) and needs to be justified, especially as it introduces an abrupt discontinuity in the ganglion cell model between 0 and nonzero temporal frequencies. Related to this point, the Robson measurements shown in Figure 4 show a marked attenuation of the CSF at low spatial frequencies for stimuli modulated at 1 Hz – such that the 1 Hz CSF is not very different from that shown for static gratings by DeValois. This suggests that the static/dynamic separation is not completely correct. Third, temporal sensitivity measured behaviorally depends strongly on spatial frequency (e.g. see Robson paper). Shouldn't this affect the argument presented in the paper – i.e. that gratings of different spatial frequencies are subject to quite different temporal filtering apparently? Some discussion of this is needed.

Unless the lack of ganglion cell responses to static images can be justified based on past experiments, the paper needs to be much more careful in asserting that known ganglion cell models can account for the discrepancy between CSFs measured behaviorally and in ganglion cells. Statements to this effect are made in many places in the paper – starting in the Abstract, Results, seventh paragraph, Discussion, fourth paragraph, etc.

*Reviewer #2:*

The authors have thoroughly revised the paper and, to my mind, have satisfied the reviewers' concerns and questions. The manuscript is much clearer and is now suitable for publication in *eLife*.

One small point that may further improve the clarity of the presentation:

In Figure 4 and Figure 4—figure supplement 1 results are presented linking model predictions to CSFs measured during the presentation of temporally modulated gratings. From the Results and Discussion, it's clear that the effects of drift are more apparent at low temporal frequency. It would be useful to see an array of curves from the model without drift, pinpointing the transition around a few Hz where transients from the stimulus override the effects of eye movements. At the very least, it would be useful to label the Figure 4—figure supplement 1 model lines with "drift".

*Reviewer #3:*

I thought the original submission was interesting and excellent. In the revised manuscript, the authors have done a thorough job of responding to both my comments and those of the other reviewers.

---

## [Author Response]

[Editors’ note: the author responses to the first round of peer review follow.]

All three reviewers agreed that the work was of broad interest. We also agreed that several issues needed to be strengthened before considering the paper further. These and several other issues are detailed in the individual reviews. The two most pressing issues follow:1) Recent work from some of the same authors has deepened understanding of the relation between FEM and neural coding. The relation between the present work and this past work is not as clear as it needs to be in the Introduction, and as a result the advance that the present work represents is difficult to appreciate immediately.

We have carefully revised Abstract, Introduction, and Discussion to make sure that the novel contributions of this study are clear. A detailed reply to this point is provided in our letter to reviewer 1, who raised this issue (see section “Overlap with previous work”). In brief, whereas our previous work focused on how eye movements affect visual input signals (i.e., that they result in a whitening of the input), here we combine measurements of eye movements with models of retinal ganglion cell responses to bridge the gap between neuronal mechanisms and psychophysics. We show that consideration of the influences of oculomotor transients on retinal responses provides a unifying account of a large range of experimental findings on human visual sensitivity. As we now better explain in the manuscript, this finding carries important implications.

2) At the heart of the paper is the discrepancy between physiological and psychophysical CSFs. But the ganglion cell examples in the first figure appear not to accurately reflect ganglion cell behavior since they do not include points at low frequencies (with instead a horizontal line from the lowest spatial frequency point plotted). Most ganglion cell measurements show some low frequency roll-off. A thorough justification of the properties used to represent measured ganglion cell CSFs is needed.

We apologize for this issue. The low-frequency properties of RGCs were incorrectly displayed in the original Figure 1A, as we had inadvertently plotted the Fourier transform of a 1D section of the receptive field, rather than a section of the 2D Fourier Transform. This led to the lack of a roll-off in the receptive field profile, which triggered the reviewers’ question. When correctly plotted, we find the standard roll-off in responses reported by many studies (see section on “M and P models”, in our reply to reviewer 1). We corrected this error in the resubmitted manuscript and now emphasize that the plots are not based on an extrapolation. We have also added additional material, including new Figure 1—figure supplement 1, to support the point that this response attenuation in physiological sensitivity falls far short of what is needed to account for psychophysical measurements of contrast sensitivity, unless eye movements are explicitly taken into account.

Reviewer #1:This paper describes the interaction between eye movements and spatial coding. The paper starts by describing a discrepancy between spatial contrast sensitivity as measured behaviorally and as measured in responses of retinal or geniculate cells. It then proposes that this discrepancy originates because the neural recordings were made in the absence of eye movements, and that the discrepancy can be resolved when eye movements are taken into account. The case for the importance of eye movements, made here and in previous work from some of the same authors, is compelling. There are several issues with the present paper, however, that limit my enthusiasm:1) M and P cell models. The literature contains numerous measurements of M and P cell spatial contrast sensitivity. From a scan of these papers (e.g. Derrington and Lennie, 1984; Kaplan and Shapley, 1982), measured spatial contrast sensitivity curves for both M and P cells fall, sometimes sharply, at low spatial frequencies. This fall is not evident in the responses depicted in Figure 1. Even the Cronor and Kaplan (1995) paper cited as the basis of Figure 1 shows a clear falloff at low spatial frequencies, and describes properties of the associated surround (related, the Benardete and Kaplan, 1997 reference in the text is about M sequences and temporal properties, so not immediately clear how it is relevant).

As mentioned above, we inadvertently plotted the 1D Fourier Transform of a section of the receptive field, rather than a section of the 2D Fourier Transform. This made the curves deviate from the receptive fields that were actually used in the simulations and appear considerably flatter at low spatial frequencies. When properly plotted (as in the revised Figure 1A), the expected low-frequency attenuation is seen. Crucially, as we show in the main text and Figure 1—figure supplement 1, this attenuation is not nearly enough to account for psychophysical CSFs. We apologize for this error and the confusion that it generated.

I am not clear on why the red curves in Figure 1 should be extrapolated from their peak to a 0 spatial frequency asymptote; this extrapolation appears to be inconsistent with most experimental results.

We now clarify that these are not extrapolations; they are the values determined by the receptive fields models measured by Croner and Kaplan (1995). As we now mention in the text, these curves were determined by the difference-of-Gaussian fits of these authors, based on measurements at spatial frequencies down to 0.07 cycles/deg, which goes beyond the range plotted. We have modified both the legend and the caption to avoid possible misunderstandings.

More generally, I think it is critical that the paper contains a thorough summary of the measured M and P properties across studies, and a clear justification of the M and P models used. Figure 1 presently undermines confidence in the central motivation for the paper.

We used the models of retinal ganglion cells (RGCs) reported by Croner and Kaplan (1995) because this study provides one of the most thorough investigations of receptive fields parameters that we could find in the literature. Unlike the few “typical” receptive fields reported by other articles, this study contains detailed lists of model parameters estimated over sizeable neuronal populations.

The degree of roll-off at low spatial frequencies reported by Croner and Kaplan is very similar to that found by other studies in macaques. See, for example, Figure 9 in Kaplan and Shapley, 1982, Figure 6 in Hicks et al., 1983, Figure 3 in Derrington and Lennie, 1984. These studies did not report parameters that we could use for modeling cell responses, except for Derrington and Lennie, who reported the spatial receptive fields’ parameters of the six P cells their Figure 3 (Table 1 in Derrington and Lennie, 1984). We have now included in the manuscript also the spatial sensitivity estimated from these parameters, which is very similar to those obtained with the data from Croner and Kaplan (Figure 1—figure supplement 1B in the resubmitted manuscript). Crucially, although the response attenuation exhibited by neurons in the low spatial frequency range is in the same direction of the attenuation in contrast sensitivity measured in humans, it falls far short of accounting for the attenuation in psychophysically-measured CSFs. In addition to the difference in the amount of attenuation, there is also a difference in the shape of the curves – a leveling-off for physiological sensitivities, compared to psychophysical measurements. These elements are clearly visible in all the figures listed above and in the new data in Figure 1—figure supplement 1.

Following the reviewer’s comments, we revised the Introduction to make clear that the deviation between neural and psychophysical measurements requires additional mechanisms. As we now discuss in the first paragraph of the Results, the only way in which Croner and Kaplan’s models could be modified to match contrast sensitivity functions is by doubling the ratio between the area of the center and surround – from the measured 0.5-0.6 values to close to 1. That is, receptive fields with highly unrealistic characteristics would be necessary to match behavioral functions. We have added a figure (Figure 1—figure supplement 1) to explain this point. Since the effect in our model originates from the redistribution of power resulting from eye movements, rather than the specific shape of the spatial receptive field of modeled neurons, our results are instead very robust.

“related, the Benardete and Kaplan, 1997 reference in the text is about M sequences and temporal properties, so not immediately clear how it is relevant.”

We thank the reviewer for pointing out this issue. The relevant article here is Croner and Kaplan, (1995). We corrected the reference in the revised manuscript.

2) Overlap with previous work. The content of Figures 1 and 2 is a review of previous work. Of particular importance, previous work from some of the same authors has established that eye movements decrease responses to low spatial frequencies in natural images. I am concerned, given that, about the intellectual advance of the present paper.

We extensively modified both the Introduction and the Discussion to clarify the significance of the present study and better explain how it goes beyond our previous work. In brief, our previous studies focused on the consequences of eye movements for visual input signals, while here, we build on these previous results to bridge between neuronal mechanisms and psychophysics. We show that standard models of retinal neurons *quantitatively* account for the way human contrast sensitivity depends on both spatial and temporal frequency, *but only* when one takes into account the temporal sensitivity of ganglion cells and their interaction with oculomotor luminance modulations. These results challenge widely accepted hypotheses about retinal functions (Atick and Redlich, 1992), which rely on the assumption that spatial filtering is sufficient to account for the shape of the CSF, and show that the proposal that space is encoded via oculomotor transients provides a unifying account of a large range of experimental findings on human visual sensitivity. As we describe in the revised manuscript, these conclusions have important consequences at the neural, perceptual, and clinical levels (Discussion).

Specifically, I think the paper would benefit from more experimental tests of the proposal. For example, are there measurements of spatial contrast sensitivity in awake behaving monkeys (i.e. with eye movements)? Recognizing that eye movements cannot be fully suppressed, can they still be manipulated to test the proposal (e.g. can they be increased)? Are there differences across eccentricity that could be exploited to provide additional tests?

While our framework makes novel predictions, we think that including further experimental tests would not improve the focus of the manuscript, as our goal here is to show that there is a logical explanation for a gap between well-established sets of measurements across many labs. However, we fully agree with the reviewer that experiments with controlled retinal image motion at selected eccentricities are highly interesting, and we have been working in this direction. We recently reported preliminary results consistent with the predictions of this work at the recent annual meeting of the Vision Sciences Society (Intoy et al., 2018). Given the reviewer’s comment, we expanded the Discussion to mention the improvements in word and object recognition reported in patients with central visual loss, when images or text are jittered or scrolled (Gustafsson and Inde, 2004; Watson et al., 2012; Harvey and Walker, 2014). These results are consistent with our prediction that a larger fixational instability should enhance sensitivity at low spatial frequencies, because of larger power available in this range (Figure 5A).

Finally, contrast sensitivity measurements have been performed in awake behaving monkeys (De Valois et al., 1974), and are included in Figure 1A, so that the reader can directly compare neurophysiological and behavioral measurements in the same species. These measurements are very similar to those reported in humans (also shown in the same figure for comparison purposes).

Reviewer #2:[…] Overall, this is an expertly conducted combination of computational modeling and eye movement recording that provides an elegant solution to a long-standing problem in vision science. My comments are aimed mainly at clarifying the presentation.The paper refers to "fixational eye movements" or "FEMs" throughout, but what is meant by this is the slow drift component of fixational eye movements, and not microsaccades or oscillatory movements. A few points. First, the paper is reasonably clear on this point, but not completely and not everywhere. For example, the casual reader who only looks at the Abstract and skims the details (including the Results where the focus on drift is pointed out) might conclude that this applies to microsaccades, since they are the best-known component of FEMs. Is there a reason to avoid using the more specific term "ocular drift" rather than "FEMs" throughout, or at least, more often?

The only reason for using “fixational eye movements” was to adopt a terminology with which readers may be more familiar. But we fully agree with the reviewer that this may lead to ambiguity, and in the revised manuscript we replaced this term with “fixational drift” (or, in many cases, simply “drift”) both in the text and figures.

Second, what is the impact of microsaccades? Would these also be expected to affect the CSF or is the scale of their temporal modulation too high to affect responses? This point should be clarified. Third, if ocular drift and microsaccades have distinguishable effects on CSF, then the authors should be especially careful with the use of the term "FEMs", especially in the Discussion, where it seems that the conclusions based on ocular drift appear to be generalized to all types of fixational eye movements.

This manuscript specifically focuses on eye drift not only because its consideration is by itself sufficient to account for experimental data, but also because humans tend to suppress microsaccades during contrast sensitivity measurements (see Mostofi et al., 2016). But we agree, microsaccades are interesting and important, and we now comment on them, as summarized below.

Briefly, microsaccades and, in general, saccades redistribute the spatiotemporal power of the retinal stimulus in a highly different manner than ocular drift, providing significantly more temporal power at low spatial frequencies. This difference leads to the prediction that microsaccades and saccades should enhance visual sensitivity at low spatial frequencies. This prediction has been recently confirmed for larger saccades (see Boi et al., 2017), but not for saccades smaller than 1 degree (Mostofi et al., 2016). A possibility, discussed in Mostofi et al., 2016 is that perhaps for microsaccades, the beneficial consequences of luminance transients and the negative consequences of saccadic suppression (a reduction in sensitivity before and during saccade) may more evenly counterbalance each other. In the resubmitted manuscript, we now comment on the possible function of microsaccades and how their input reformatting differs from drift in the Discussion (fifth paragraph).

In several places – starting with the title – the manuscript implies that there is an active strategy behind these effects of FEMs, rather than it being an incidental effect. To me, "active strategy" implies some reliable relationship between the circumstance and the subject's behavior. For example, the needle-threading task shows that microsaccades appear to be generated based on some sort of active strategy. For ocular drift, it is not clear that such a relationship has been established. Do you have some basis of this claim, or is it simply a speculation? If it is a speculation, the text should be written more conservatively to match.

Our use of the term “active” was simply intended to convey the notion that contrast sensitivity is not merely the outcome of sensory processes, but it also includes oculomotor contributions. To avoid ambiguity, we changed the title by replacing the word “active” with “oculomotor” and revised the text similarly. However, there is some evidence that humans do in fact actively control the overall amount of drift, and we now comment on this in the Discussion (eighth paragraph) in the context of the predictions of our study.

Another plausible explanation for why the behavioral CSF does not match the properties of the retinal output is that some part of the rest of the visual system is responsible. Is there a basis for excluding this possibility?

This is a reasonable possibility that we cannot exclude; we added a comment about this in the fourth paragraph of the Discussion. However, while one might expect contributions from multiple stages of the visual system, our results show that retinal sensitivity and fixational drift suffice to account for the CSF over a broad spatiotemporal range, without a further downstream reshaping. This is a further reason why we think our results are significant, as we also now comment.

Reviewer #3:This paper explores how non-separable spatio-temporal frequency tuning of M and P cells combines with measured fixational eye movements to account for observed behaviorally measured visual contrast sensitivity. A minimal model agrees well with a range of experimental data (free viewing of gratings at different temporal frequencies as well as retinal stabilization experiments) and suggests that eye movements have been optimized for neuronal contrast sensitivity curves (this work) as well as the content of natural scenes (previous work from the same groups). Experimental tests are suggested, particularly for retina-in-a-dish style experiments, where movements typical of FEMs are rarely added to visual stimuli. Experiments are also suggested for human subjects and for some clinical applications.1) The paper is clearly presented and the work is thorough and interesting, but there should be more emphasis in the Introduction on why it's so important to understand the discrepancies between neuronal and behavioral CSFs for stationary stimuli. This could be addressed with a bit of text that previews some of the Discussion points about the neural coding and potential clinical applications of these findings.

We rewrote large sections of the Introduction to better explain the rationale for our work. As suggested by the reviewer, we now start by reviewing the relation between the shape of the CSF and theories of efficient encoding and mention that an explanation of contrast sensitivity in humans has important conceptual consequences and potential clinical implications. While the discrepancies fundamentally come down to quantitative measurements, there are two main qualitative points about function: whether the physiologically-measured low-frequency attenuation in neural responses accounts for spatial decorrelation of natural images (as theorized by Atick and Redlich), and whether the physiologically-measured spatiotemporal properties of ganglion cell receptive fields account for the spatiotemporal inseparability present in the human CSF. In both cases, we show that the effects of FEMs – and not just neuronal filtering characteristics – are critical. The Introduction section has been modified extensively to explain these points, see the third and seventh paragraphs.

2) Some recap of existing retinal recordings with FEM-like perturbations (e.g. Greschner et al., 2002) should be added to the Discussion.

We expanded discussion of previous physiological studies in which stimuli included temporal modulations like those resulting from fixational eye movements. These studies, most of them performed in vitro, provide support to our proposal that FEMs are an integral component of the retinal neural code. The relevant text is in the seventh paragraph of the Discussion.

3) Figure 1A – no data points are apparent below about 2Hz for the P and M cells, right around where the neuronal and behavioral CSFs diverge. It would be nice to include some data points there, perhaps from other studies. If data points do exist at 0Hz, they should be made more apparent.

If we understand correctly, the reviewer is referring to 2cpd, not 2Hz, as this is what was plotted on the abscissa of Figure 1A. As mentioned above, this figure was problematic in several respects, all of which have now been fixed. Specifically, the physiological data are not extrapolations; they are the receptive fields of retinal ganglion cells given by standard difference of Gaussians models, as measured by Croner and Kaplan (1995). The symbols previously on these curves were simply markers used to distinguish the curves. We can see how they could be misinterpreted as data points, so we now distinguish the curves by line styles. The parameters for these models were experimentally estimated by Croner and Kaplan by presenting gratings at different spatial frequencies, with a minimum of 0.07 cpd (Croner and Kaplan, 1995), which extends below the range plotted.

[Editors' note: the author responses to the re-review follow.]

The revisions have improved the paper substantially. There is one outstanding point remaining about the ganglion cell model (see reviews for details). In consultation all of the reviewers agreed this was important to resolve.Reviewer #1:This is a revision of a paper about the relationship between fixational eye movements and spatial contrast sensitivity. The paper centers around the observation that the spatial contrast sensitivity function measured behaviorally in primates differs from that of retinal ganglion cells, particularly at low spatial frequencies. The paper argues that eye movements and the temporal sensitivity of ganglion cells account for this difference.The paper has improved in revision. In particular, I found the Introduction more compelling, and the connection of the spatial ganglion cell models to past work is much less confusing with Figure 1 fixed. An issue that became clearer in the revised version of the paper is that the temporal model for the ganglion cells needs to be described more thoroughly. This and some smaller points follow.The crux of the paper is how the dynamics of eye movements make nominally static spatial inputs dynamic, and this shifts static inputs into a temporal region where the ganglion cells are more responsive. The ganglion cell model in Equation 4 is taken straight from prior measurements, and hence is well supported in the literature (see suggestion below about including relevant parameters). However, this model does not predict a complete lack of response at a temporal frequency of 0 Hz, as assumed in the paper.

We think we see where some of the difficulties have arisen. It is impossible to measure responses that are truly at 0 Hz, as it would require an infinitely long experiment to do so – and this holds not only for physiological measurements, but also for psychophysical ones. As a practical matter, the lowest frequency measurable is comparable to the reciprocal of the duration of a trial (i.e., 0.2 or 0.3 Hz); any attempt to infer behavior at lower frequencies would be confounded by the visual input on adjacent trials (or between trials). With this in mind, we had intended our statements about insensitivity to 0 Hz as shorthand for the more legalistic, “negligible sensitivity below the frequencies at which sensitivity can practically be measured.” As explained in detail below, this hypothesis is well justified from the data available in the literature, and our results are robust to the specifics of how exactly this hypothesis is incorporated into the neural models. We now make our point of view explicit in the paper and better explain how the model relates to physiological data at low temporal frequencies.

There are several issues with how this is treated in the paper. First, the temporal model is central to the paper, and should not be relegated entirely to the Materials and methods (one suggestion would be to add a figure between the current Figure 2 and 3 showing the temporal response of the ganglion cell model).

We fully agree with the reviewer and added a panel in Figure 2. This new panel (Figure 2E) shows the temporal profiles of both M and P cells. We also provide two new tables (Table 1 and 2), in which we report the values of the parameters used to model both spatial (Table 1) and temporal kernels (Table 2).

Second, is there experimental evidence that static components of the stimulus do not modulate the ganglion cell responses? This assumption should be clarified earlier (in reference to Figure 3) and needs to be justified, especially as it introduces an abrupt discontinuity in the ganglion cell model between 0 and nonzero temporal frequencies.

Strictly speaking, 0 Hz is a mathematical abstraction, and measuring sensitivity at this frequency is not physically possible. We therefore focus here on the broader question of sensitivity of retinal ganglion cells to low temporal frequencies, an issue that regards primarily P cells. These are the neurons with more sustained responses (Figure 2E). There is considerable experimental evidence in support of the way we handle the low-frequency limit, and we have added two paragraphs to the Discussion as well as a figure (Figure 4—figure supplement 3) to comment on this issue.

It is first important to realize that, in most neurophysiological (and psychophysical) studies, the data reported in the low temporal frequency range do not provide reliable estimation of sensitivity. This happens for multiple reasons, including: the too short duration of the experimental trial; the lack of consideration of the visual stimuli present before and after each trial, and the length of the estimated impulse response.

Typically, the transfer functions reported at low temporal frequencies are extrapolations outside of the range of measured values based on models that were not designed for this purpose (e.g., the linear cascade model (Victor, 1987) in Benardete and Kaplan, 1997 and 1999; a difference of exponential in Derrington and Lennie, 1984). Both of these models use functional forms that flatten out at very low temporal frequencies, but this flattening occurs below the frequencies at which data are acquired to fit the model. These extrapolations must be interpreted with great caution, as they merely reflect untested model assumptions. Victor’s (1987) linear cascade model estimated by Kaplan and colleagues, which we used in our study (Figure 2E), was never meant to serve as an extrapolation to frequencies outside of the range used to fit it, and this is why we don’t merely use model values down to DC. Benardete and Kaplan, for example, only measured impulse responses for ~0.5 sec, so the frequencies in the Fourier transform of the impulse response, which they used to fit the model, did not go below 2 Hz.

So, rather than use these extrapolations, we turn to the very few studies that specifically examined low temporal frequencies in retinal ganglion cells. These studies found a decline in sensitivity up to the limit that they could measure. This applies both to the more sustained X channel in the cat (Frishman et al., 1987; Victor, 1987), as well as P cells in the macaque, as shown in Figure 12A-B in Purpura et al., 1990. These studies suggest that the response attenuation takes the form of an approximately linear decrease in loglog scale. Such behavior is also expected from theoretical considerations based on the characteristics of adaptation (Thorson and Biederman-Thorson, 1974), considerations that seem to apply to the responses of cones in the retina of the macaque (Boynton and Whitten, 1970) and therefore will limit the low-frequency behavior of retinal ganglion cells.

Independently, any retinal sensitivity at frequencies ~0.3 Hz and below is likely to be virtually useless in a psychophysical experiment in trials of 2-3 sec or less – because whatever low-TF signal is present is likely to be masked by low-TF noise contributed by visual input on adjacent trials, or what the subject does between trials (e.g., looks around the lab, blinks, etc.).

Our model is highly robust to the specifics of how this reduction in sensitivity at low temporal frequencies is implemented in the simulations. In the manuscript, we simply discarded responses below a frequency threshold of 0.63Hz. But results were virtually identical when we used different frequency thresholds, or when we modeled sensitivity as a power law function of temporal frequency in the low-frequency range (as in Purpura et al., 1990, and Thorson and Biederman-Thorson, 1974, respectively).In the latter case, results are also robust with respect to the slope of the power-law function. We comment on these results in the Discussion and refer the interested reader to a new supplementary figure (Figure 4—figure supplement 3).

Related to this point, the Robson measurements shown in Figure 4 show a marked attenuation of the CSF at low spatial frequencies for stimuli modulated at 1 Hz – such that the 1 Hz CSF is not very different from that shown for static gratings by DeValois. This suggests that the static/dynamic separation is not completely correct.

We do not follow the reviewer here: our model correctly predicts the gradual change in spatial sensitivity shown in the Robson data, as the temporal frequency of the grating is increased (see Figure 4). There is no abrupt transition in the predicted CSF, which is consistent with the smooth transition between dynamic and static power in the retinal input (see Figure 2C).

Perhaps, confusion occurred here between the temporal modulation of the stimulus on the monitor (0 Hz for a static grating) and the temporal fluctuations in the responses of our models (where 0 Hz indicates a constant response). The two things are not equivalent: we always use model responses at all non-zero temporal frequencies – rather than just at the temporal frequency of the grating – to estimate contrast sensitivity. We think part of this confusion was generated by the previous version of Figure 2, which was giving the false impression of a dichotomy in the visual input by showing next to each other the spatial distributions of power on the retina at 0 Hz and the power integrated across all non-zero temporal frequencies (panels D and E respectively). These two curves were only meant to show that the power distributions differ at different temporal frequencies (already evident from the map in Figure 2B), not to imply the presence of a discontinuity.

In the revised manuscript, we have modified Figure 2 to eliminate possible ambiguity. We also added a supplementary figure (Figure 4—figure supplement 2) in which we show the predicted CSF at temporal modulations of the gratings not present in the Robson data in Figure 4, so to further highlight that the model well captures the smooth low-pass to band-pass transition in the CSF.

Third, temporal sensitivity measured behaviorally depends strongly on spatial frequency (e.g. see Robson paper). Shouldn't this affect the argument presented in the paper – i.e. that gratings of different spatial frequencies are subject to quite different temporal filtering apparently? Some discussion of this is needed.

A strength of our study is that it explains the way human temporal sensitivity varies across spatial frequencies (a space-time inseparable function) on the basis of space-time separable neural filters, like the ones of retinal ganglion cells. In other words, our model captures the full spatiotemporal pattern of contrast sensitivity by means of a linear combination of the sensitivity of P and M cells, without requiring additional temporal filters to process different ranges of spatial frequencies, as one may think. We have added a paragraph to the Discussion to comment on this point.

Unless the lack of ganglion cell responses to static images can be justified based on past experiments, the paper needs to be much more careful in asserting that known ganglion cell models can account for the discrepancy between CSFs measured behaviorally and in ganglion cells. Statements to this effect are made in many places in the paper – starting in the Abstract, Results, seventh paragraph, Discussion, fourth paragraph, etc.

As explained above, our model is well justified by the literature. We have carefully revised the text, including the points highlighted by the reviewer, to eliminate confusion and make sure that our claims are clear.

Reviewer #2:The authors have thoroughly revised the paper and, to my mind, have satisfied the reviewers' concerns and questions. The manuscript is much clearer and is now suitable for publication in eLife.One small point that may further improve the clarity of the presentation:In Figure 4 and Figure 4—figure supplement 1, results are presented linking model predictions to CSFs measured during the presentation of temporally modulated gratings. From the Results and Discussion, it's clear that the effects of drift are more apparent at low temporal frequency. It would be useful to see an array of curves from the model without drift, pinpointing the transition around a few Hz where transients from the stimulus override the effects of eye movements. At the very least, it would be useful to label the Figure 4—figure supplement 1 model lines with "drift".

We thank reviewer 2 for the nice comments. We added the suggested new figure (Figure 4—figure supplement 2), which plots the CSF predicted by our model at two additional intermediate temporal modulating frequencies: 2 Hz and 3 Hz. This figure shows that the transition from a band- to a low-pass behavior of the CSF occurs around 3 Hz. This result is consistent with psychophysical results; see, for example, the data from Bowker and Tulunay-Keesey, 1983, reported in their Figure 1, as we now mention in the eleventh paragraph of the Results section. We also re-labelled the curves in Figure 4—figure supplement 1 and Figure 5—figure supplement 1 with “Drift”, as suggested by the reviewer.